# Assessment of the Prescriptions of Systemic Antibiotics in Primary Dental Care in Germany from 2017 to 2021: A Longitudinal Drug Utilization Study

**DOI:** 10.3390/antibiotics11121723

**Published:** 2022-11-30

**Authors:** Gabriele Gradl, Marita Kieble, Jens Nagaba, Martin Schulz

**Affiliations:** 1German Institute for Drug Use Evaluation (DAPI), 10557 Berlin, Germany; 2German Dental Association (BZÄK), 10115 Berlin, Germany; 3Department of Medicine, ABDA–Federal Union of German Associations of Pharmacists, 10557 Berlin, Germany; 4Institute of Pharmacy, Freie Universität Berlin, 12169 Berlin, Germany

**Keywords:** anti-bacterial agents, clindamycin, drug utilization, dental care, primary health care, insurance claims analysis

## Abstract

(1) Background: Due to increasing antibiotic resistance, the frequency of antibiotic use should be questioned in dentistry and attention paid to the choice of the best suited substance according to guidelines. In Germany, overprescribing of clindamycin was noteworthy in the past. Therefore, the aim of our study was to determine the trend of antibiotic prescriptions in primary dental care. (2) Methods: Prescriptions of antibiotics in German primary dental care from 2017 to 2021 were analysed using dispensing data from community pharmacies, claimed to the statutory health insurance (SHI) funds, and compared with all antibiotic prescriptions in primary care. Prescriptions were analysed based on defined daily doses per 1000 SHI-insured persons per day (DID). (3) Results: Amoxicillin was the most frequently prescribed antibiotic (0.505 DID in 2017, 0.627 in 2021, +24.2%) in primary dental care, followed by clindamycin (0.374 DID in 2017, 0.294 in 2021, −21.4%). Dental prescriptions still made up 56% of all clindamycin prescriptions in primary care in 2021. (4) Conclusions: Our study suggests that the problem of overuse of clindamycin in German dentistry has improved, but still persists.

## 1. Introduction

Antimicrobial resistance, driven by misuse and overuse of antibiotics, is a major and global public health challenge [1]. Therefore, promoting the prudent use of antibiotics, as well as communication to healthcare professionals in this regard, is a part of the European Commission’s Pharmaceutical Strategy for Europe [2].

The consumption of antibacterials for systemic use is substantially higher in the community compared to the hospital sector. Ninety one percent of antibacterial use, reported by the European Union (EU) Member States and European Economic Area (EEA) countries for 2020, occurred in the community [3]. Dental prescriptions accounted for a large proportion of systemic antibiotic use in the community over the past decade. In Belgium, Scotland, Sweden, Norway, and England, for example, this proportion was between 6% and 8%, and in Germany, it was 9% between 2012 and 2015 [4,5,6]. With a view to bacterial resistance, it is, therefore, important to monitor dental antibiotic prescribing in primary care in order to identify potential to reduce unnecessary antibiotic prescriptions. In dentistry, the use of antibiotics is indicated for prophylaxis of infective endocarditis or the treatment of certain odontogenic infections [7,8]. In localized infections without a tendency to spread, for example, antibiotic therapy should be avoided altogether and causal therapy should always be the focus [9].

Selection of the appropriate antibiotic appears to be another issue in dentistry where there is potential for improvement [6,10,11,12]. The antibiotics to be used primarily for odontogenic infections, according to the German dentistry guideline, are aminopenicillins, if necessary in combination with a beta-lactamase inhibitor [9]. The prescription of clindamycin, on the other hand, is recommended only for patients with a penicillin allergy. The practice of antimicrobial prescribing for dental and oral infections in primary care differs between European countries [5,6,13]. For example, in Germany in 2016 amoxicillin was prescribed most frequently, followed by clindamycin [13]. Although clindamycin is not the antibiotic of first choice for the treatment of odontogenic infections, according to the guideline, it was used very frequently in Germany in the past [13,14,15,16,17]. In Belgium, amoxicillin without or with a beta-lactamase inhibitor was prescribed most frequently by dentists, followed by clindamycin [6]. Clindamycin was also ranked third in Sweden and Norway–after phenoxymethylpenicillin and amoxicillin–and sixth in England and Scotland–after amoxicillin, metronidazole, erythromycin, doxycycline and phenoxymethylpenicillin [5].

In Germany, data are scarce on the outpatient prescription of antibiotic agents by dentists and current figures are lacking [4,11,12]. The extent to which the use of clindamycin, in particular, has changed in outpatient dentistry in recent years has not yet been quantitatively examined. The aim of our study was to use dispensing data from community pharmacies to analyse the antibiotic prescribing behaviour of dentists over the past five years, i.e., which substances were prescribed and in what quantities. Specifically, we wanted to determine to what extent dental clindamycin prescriptions were changing.

## 2. Results

### 2.1. Dental Prescriptions by Year and Substance

Dental prescriptions of systemic antibiotics in primary care remained stable over the period 2017 to 2021 (1.20–1.25 DID). Dental prescriptions as a percentage of all antibiotic prescriptions increased from 10% to 15% during this period. The most commonly dispensed antibiotics in 2021 were amoxicillin (0.63 DID, 50% share), clindamycin (0.29 DID, 24% share), amoxicillin and beta-lactamase inhibitor (0.16 DID, 13% share), phenoxymethylpenicillin (0.08 DID, 6% share), doxycycline (0.03 DID, 2% share), cefuroxime (0.02 DID, 2% share), and metronidazole (0.01 DID, 1% share). Prescriptions of clindamycin declined by 21% over the period, from 0.37 DID in 2017 to 0.29 in 2021. Prescriptions of phenoxymethylpenicillin also decreased (0.11 DID in 2017, 0.08 in 2021, −32%). In contrast, the share of prescriptions of amoxicillin without a beta-lactamase inhibitor (0.51 DID in 2017, 0.63 DID in 2021, +24%) and with a beta-lactamase inhibitor (0.10 DID in 2017, 0.16 DID in 2021, +63%) increased (Table 1).

### 2.2. All Antibiotic Prescriptions by Year and Substance

All prescriptions of systemic antibiotics in primary care decreased by –36%, from 12.59 DID in 2017 to 8.06 in 2021. Prescriptions of amoxicillin (2.33 DID in 2017, 1.58 DID in 2021, –33%), and doxycycline (1.60 DID in 2017, 1.16 DID in 2021, –28%) decreased at about the same rate as all prescriptions. There was a particularly sharp decrease in prescriptions of phenoxymethylpenicillin (0.64 DID in 2017, 0.30 DID in 2021, −53%) and cefuroxime (2.14 DID in 2017, 0.99 DID in 2021, −54%), and a smaller decrease in clindamycin prescriptions (0.63 DID in 2017, 0.53 DID in 2021, −16%). In contrast, prescriptions of amoxicillin in combination with a beta-lactamase inhibitor (0.58 DID in 2017, 0.75 DID in 2021, +31%) increased (Table 2).

### 2.3. Antibiotic Prescriptions by Month

The monthly mean (± standard deviation) DID of dental prescriptions during the period was 1.20 ± 0.07. Monthly values were 1.17 in January 2017, and 1.31 in December 2021, and they ranged between 1.03 and 1.36. Thus, they were not subject to large fluctuations. Before the start of the COVID-19 pandemic, DID of all antibiotics showed typical seasonal fluctuations with high values during winter months and low values during summer months. In the period from January 2017 to March 2020, values ranged between 8.92 and 17.93 DID (mean 11.99 ± 2.27). From April 2020 to December 2021, they ranged between 6.32 and 10.86 DID (mean 7.84 ± 1.16) (Figure 1).

### 2.4. Analysis of Clindamycin Prescriptions

Prescriptions of clindamycin by dentists decreased from 0.37 DID (31% share of all dental antibiotic prescriptions) in 2017 to 0.29 DID (24% share) in 2021 (Figure 2). All clindamycin prescriptions in primary care also declined during the period, from 0.63 to 0.53 DID, but with an increasing share of all dispensed antibiotics DID (from 5.0% to 6.5%). Clindamycin DID, when prescribed by dentists as a percentage of all clindamycin prescriptions, decreased from 60% to 56% over the period.

## 3. Discussion

### 3.1. Prescriptions of All Antibiotics in Primary Dental Care

Dental prescriptions make up a substantial proportion of all antibiotics prescribed in primary care. The share of antibiotics dispensed in dental prescriptions increased from 10% to 15% for the period 2017–2021, mainly due to a decrease in all antibiotic prescriptions in primary care. This decrease in antibiotic use could be explained, in part, by decreased prescriptions for respiratory diseases for children [18]. A sharp fall in prescriptions for amoxicillin, azithromycin, and cefuroxime in 2020 suggested that this was due to COVID-19 pandemic-related measures of hygiene and the corresponding decrease in the occurrence of respiratory tract infections [19]. In accordance with this assumption our current analysis of monthly antibiotic prescriptions showed that, during the COVID-19 pandemic in the months beginning from March 2020, the monthly fluctuations in DID, as well as the total number of prescriptions, decreased considerably compared to the period before March 2020. In contrast, dental antibiotic prescriptions did not change substantially during the study period. The influence of nationwide restrictions on public and social life at the end of March 2020 on dental antibiotic prescription DID could not be detected. This observation was not expected, since across England, for example, the restricted access to dental care due to the COVID-19 pandemic increased antibiotic prescribing, such that antibiotic prescribing in April to July 2020 was 25% higher than April to July 2019, with a peak in June 2020 [20]. Assuming that the morbidities of patients in German dentistry are not substantially different, we may, therefore, conclude that there was no substantial impact by the COVID-19 pandemic on antibiotic prescribing.

In order to assess whether there could have been potential for savings in dental prescriptions, one would need to estimate whether there were non-indexed prescriptions and what the proportion of them might have been. A study in the United Kingdom has shown that only 19% of antibiotic prescribing in general dental practice in 2016 was carried out according to guidelines [21]. Studies on primary dental care across a number of different countries identified factors such as clinical time pressure, delay of treatment, refusal of operative treatment, and also unawareness, or lack, of respective guidelines as being associated with (inappropriate) antibiotic prescribing [22,23,24].

Very few comprehensive studies exist on the antibiotic prescribing behaviour of German dentists [25]. An educational intervention study on the prescribing behaviour of dentists in the German federal state of Mecklenburg West Pomerania in 2013, found that there might be different reasons for non-indicated prescriptions of antibiotics [26]. Time pressure before weekends and holidays, as well as in emergency service, was one reason. Uncertainty in the treatment of patients with cardiac diseases, and the possible need for antibiotic prophylaxis for infective endocarditis, was another. Moreover, pressure to meet the patient’s desire for rapid pain relief was cited as a further reason.

### 3.2. Prescriptions of Clindamycin in Primary Dental Care

The focus of antibiotic therapy for odontogenic infections is to use the most effective and safe substance [7,9]. Data from the German Antimicrobial-Resistance-Surveillance (ARS) database shows that the most common pathogens in German dental practices are *Streptococcus* spp. and *Staphylococcus* spp. and that clindamycin had comparatively high resistance rates regarding these (17–19%) [27]. Penicillins showed good efficacy in the treatment of odontogenic infections with these bacteria [9]. In contrast, clindamycin is well known for the highest rate of both fatal and non-fatal adverse drug reactions (ADR) of all antibiotics commonly prescribed by dentists [28]. Gastrointestinal symptoms are common with the use of clindamycin, and, from 2010–2018, clindamycin-related ADR on the gastrointestinal tract, skin, cardiovascular system, and central nervous system were frequently reported to the Drug Commission of German Dentists (Arzneimittelkommission Zahnärzte; AKZ) [29,30,31,32,33,34]. Until 2015, clindamycin led the statistics of spontaneously reported ADR to the AKZ. Case numbers declined for the first time in 2016, but rose again the following year. Compared to aminopenicillins, side effects, including pseudomembranous colitis, caused by community-associated *Clostridium difficile* infection, occurred more frequently with this antibiotic [28,35]. Therefore, the AKZ advises the use of a beta-lactam antibiotic (with a beta-lactamase inhibitor if necessary) as a first-line antibiotic and only resorting to clindamycin as a reserve, e.g., in cases of penicillin allergy [7,36].

Past studies revealed overprescribing of clindamycin in German dental care and markedly higher use of this antibiotic compared to other European countries, such as England, Scotland, Norway and Sweden [5,12]. Our study showed that in primary dental care amoxicillin was the most commonly used antibiotic. Its prescription increased by 24% over the period and its proportion by 8 percentage points. Although the proportion of clindamycin prescriptions decreased at similar rates, it was still the second most commonly prescribed antibiotic by dentists. These high prescription rates cannot be explained solely by necessary treatments of patients with penicillin allergy. For example, it was found that the anamnestic indication of a suspected penicillin allergy exceeded the frequency of confirmed cases. While 4.5% of persons in southern Europe and 10% in the USA reported suffering from penicillin allergy, most of them might, in fact, be nonallergic [37,38]. In German dental practice, testing for penicillin allergy is rather rare. Thus, it may be assumed that clindamycin is not always used according to the guideline. Together with the frequency of clindamycin related ADRs this underpins the need for action with respect to a reduction in dental clindamycin prescriptions.

The Federal Joint Committee (G-BA), the highest decision-making body of the joint self-government of physicians, dentists, hospitals and health insurance funds in Germany, issues directives for the benefit catalogue of the SHI funds and, thus, specifies which services in medical care are reimbursed. The G-BA has planned a new guideline on antibiotic care in dentistry for 2023. As one aspect of the guideline, recommendations are planned for the prescription of clindamycin. For this purpose, the German Dental Association (BZÄK) works in an advisory capacity to the G-BA working group for the quality assurance procedures for systemic antibiotic therapy in dental treatment [39]. We hope that implementation of the new guideline by regional dental associations and regional health insurance funds provides further impetus to reducing the use of clindamycin. We recommend conducting intervention studies, focusing on optimising dental antibiotic prescribing in Germany, since such studies have shown encouraging results [40,41]. In addition, we would like to emphasise the importance of continuous training for dentists through publications and regular participation in continuing education courses [36,42,43].

### 3.3. Prescriptions of Other Antibiotics in Primary Dental Care

The overall balance in the trend of prescriptions was mainly caused by a decrease in prescriptions for the antibiotics clindamycin (−0.08 DID), phenoxymethylpenicillin (−0.04 DID), doxycycline (−0.01 DID) and cefuroxime (−0.01 DID), and an increase in prescriptions for amoxicillin, both without (+0.12 DID) and with a beta-lactamase inhibitor (+0.06 DID).

The aminopenicillin amoxicillin is effective as a broad-spectrum antibiotic against both Gram-positive and Gram-negative bacteria and is recommended as a first-line antibiotic, according to the German dentistry guideline [9]. The narrow-spectrum antibiotic phenoxymethylpenicillin is not recommended by the German Society for Dental, Oral and Maxillofacial Medicine (Deutsche Gesellschaft für Zahn-, Mund- und Kieferheilkunde e. V.; DGZMK) as first-line therapy for dentogenic infections, which may account for the declining prescription rate of this antibiotic during the study period [44]. This beta-lactam antibiotic was also prescribed less frequently than amoxicillin in primary dental care in Belgium, England and Scotland, but was used more frequently than aminopenicillins in Norway and Sweden [5,6]. Different recommendations for the use of antibiotics in dentistry in these European countries are the probable causes of the observed differences [5].

Cefuroxime is still prescribed too frequently in primary care in Germany although this antibiotic is on the World Health Organization’s ‘watch group antibiotics’ list [45,46,47]. As a second-generation cephalosporin this antibiotic has the potential of selecting extended-spectrum beta-lactamase-producing Enterobacteriaceae, as well as increasing the risk of *Clostridium difficile* infections [35,48]. The decrease in prescriptions could, therefore, be explained in terms of increased consideration of these risks [9]. In Belgium, only about one fifth of cefuroxime DID were prescribed in primary dental care, compared to Germany, while this antibiotic seems to be of no importance in England, Scotland Norway and Sweden [5,6].

The decline in prescriptions of doxycycline might also be related to a high rate of resistance to this antibiotic [49,50]. In Belgium, England, Scotland, Norway and Sweden, dental prescriptions of doxycycline also declined in the period from 2014 to 2016 [5,6].

The increase in prescriptions of amoxicillin with a beta-lactamase inhibitor should be viewed critically. Increased use of this combination carries the risk of antimicrobial resistance [51]. In addition, this antibiotic is also potentially hepatotoxic [52,53]. An increase in dental prescriptions of this antibiotic was also observed in Belgium in the period from 2014 to 2016 [6].

### 3.4. Strengths and Limitations

The major strength of this study is that it was based on dispensing data from the vast majority of community pharmacies in all federal states of Germany and, thus, was a highly representative sample, representing 88% of Germany’s population. Only data from privately insured patients and for services not charged to the SHI system were not available in the database. The proportion of privately insured persons in Germany averaged 11% during the analysis period [54]. Another strength is that the database, being constantly updated, allowed us to cover the most recent period in our study.

A limitation is that neither patient data nor data on indication e.g., type of odontogenic infection, were available in our database. Therefore, we could not evaluate the quality of dental antibiotic prescribing, in general, or the appropriateness of the use of clindamycin, in particular. Furthermore, the use of DID to measure antibiotic prescribing may be unsuitable for drugs used in children, because in this patient group dosing can be very different from that for adults.

## 4. Materials and Methods

### 4.1. Study Design

We conducted a longitudinal drug utilization study in the period 2017–2021, querying the database of the German Institute for Drug Use Evaluation (DAPI) [55]. This database contains anonymous dispensing data from community pharmacies claimed to the statutory health insurance (SHI) funds. The SHI system, consisting of nearly 100 funds, covers 88% of the population, i.e., approximately 73.3 million people. Claims data from a representative sample of more than 80% (until June 2019) and more than 95% (from July 2019 onwards) of the community pharmacies in all 16 federal states were available. Data were extrapolated by regional factors to 100% of the SHI-insured population [46]. Prescriptions for privately insured patients were not covered by the database. Data on the indication, treatment duration, or dosages, as well as patient characteristics, were also not available.

### 4.2. Classification of Data and Measurement of Antibiotic Prescriptions

Dental prescriptions could be differentiated in the database from medical antibiotic prescriptions in primary care by different identification features of the prescriptions. In the group “All prescriptions” dental and medical prescriptions were included.

Dispensing data were linked to a database containing information on the name, composition, active ingredients, package size, dosage form, and route of administration using the specific product code (Pharmazentralnummer, an identification number for pharmaceutical products in Germany). We classified antibiotics according to the official version of the Anatomical Therapeutic Chemical (ATC) classification system of the World Health Organization Collaborating Centre for Drug Statistics Methodology [56]. Defined daily doses (DDD) were obtained from the official version of the German ATC-Classification [57]. In general, the DDD was the assumed average daily maintenance dose for the main indication of a drug in adults. We included antibiotics for systemic use, which fell into the ATC codes J01 and P01AB01 (metronidazole).

Antibiotic use was estimated by defined daily doses per 1000 SHI-insured persons per day (DID) on an annual and monthly basis [45]. The number for SHI-insured persons was obtained from the Federal Ministry of Health [58]. Dental antibiotic prescriptions, all antibiotic prescriptions, and the share of the antibiotic substances in relation to all dental antibiotic prescriptions and to all antibiotic prescriptions were calculated. The substances doxycycline (J01AA02), amoxicillin (J01CA04), phenoxymethylpenicillin (J01CE02), amoxicillin and beta-lactamase inhibitor (J01CR02), cefuroxime (J01DC02), clindamycin (J01FF01) and metronidazole (P01AB01) were analysed separately. All other substances were evaluated collectively as other antibiotics.

Antibiotic prescriptions by dentists in Germany from 2017 to 2021 were analysed and compared with all antibiotic prescriptions in primary care. The substances amoxicillin, clindamycin, amoxicillin and beta-lactamase inhibitor, phenoxymethylpenicillin, doxycycline, cefuroxime, and metronidazole, prescriptions of each of which accounted for ≥1% of DID of all dental antibiotic prescriptions, were selected for analysis per year. In order to check for potential influences of the COVID-19 pandemic, dental and all antibiotic prescriptions in primary care were also subjected to a detailed analysis per month.

## 5. Conclusions

An increasing proportion of antibiotic prescriptions in German primary dental care are beta-lactam antibiotics, in accordance with the first line recommendations of the German dentistry guidelines for many odontogenic infections. The downward trend in dental clindamycin prescriptions in recent years is encouraging. Nevertheless, it should be noted that the use of this antibiotic is still very high in Germany, compared to other European countries. However, since no information on patient characteristics, including diagnoses, was available in the database used, it must be taken into account that specific reasons and potential interventions for reducing dental antibiotic prescriptions remain to be explored. Hence, further studies are needed to shed light on this.

## Figures and Tables

**Figure 1 antibiotics-11-01723-f001:**
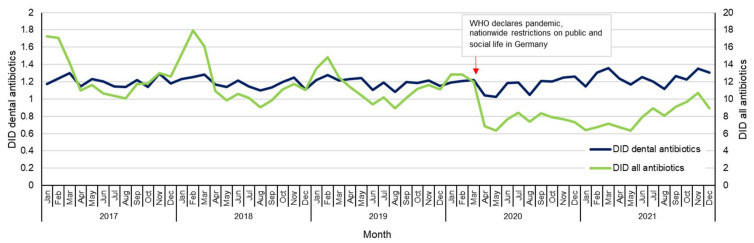
Prescription of systemic antibiotics in dental and primary care overall from January 2017 to De-cember 2021 in DID. Abbreviations: DID, defined daily doses per 1000 statutory health-insured persons per day; WHO, World Health Organization.

**Figure 2 antibiotics-11-01723-f002:**
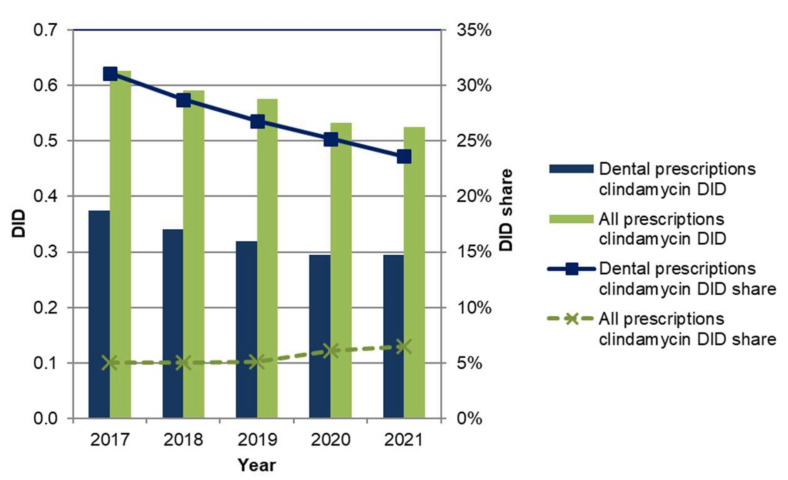
Trends for dental and all clindamycin prescriptions in primary care.

**Table 1 antibiotics-11-01723-t001:** Prescription of systemic antibiotics by dentists during 2017 to 2021 in DID. Abbreviations: DID, defined daily doses per 1000 statutory health-insured persons per day; %, share of all prescribed antibiotics; Δ, difference between 2021 and 2017 DID; Δ %, percentage difference between 2021 and 2017 DID.

Year	2017		2018		2019		2020		2021			
Substance	DID	%	DID	%	DID	%	DID	%	DID	%	Δ	Δ%
All antibiotics	1.200	100.0%	1.185	100.0%	1.193	100.0%	1.169	100.0%	1.245	100.0%	0.045	3.7%
Amoxicillin	0.505	42.1%	0.530	44.7%	0.560	47.0%	0.570	48.7%	0.627	50.4%	0.122	24.2%
Clindamycin	0.374	31.1%	0.340	28.7%	0.319	26.8%	0.295	25.2%	0.294	23.6%	−0.080	−21.4%
Amoxicillin and beta-lactamase inhibitor	0.100	8.4%	0.113	9.5%	0.128	10.8%	0.139	11.9%	0.164	13.1%	0.064	62.8%
Phenoxymethylpenicillin	0.110	9.2%	0.099	8.3%	0.090	7.5%	0.079	6.8%	0.075	6.0%	−0.035	−32.1%
Doxycycline	0.036	3.0%	0.032	2.7%	0.030	2.5%	0.027	2.3%	0.026	2.1%	−0.010	−26.2%
Cefuroxime	0.028	2.3%	0.026	2.2%	0.024	2.0%	0.022	1.9%	0.022	1.8%	−0.006	−20.7%
Metronidazole	0.015	1.3%	0.015	1.3%	0.016	1.3%	0.014	1.2%	0.014	1.1%	−0.001	−8.9%
Other antibiotics	0.032	2.6%	0.030	2.6%	0.026	2.1%	0.023	2.0%	0.023	1.9%	−0.009	–28.7%

**Table 2 antibiotics-11-01723-t002:** Prescription of systemic antibiotics in primary care during 2017 to 2021 in DID. Abbreviations: DID, defined daily doses per 1000 statutory health-insured persons per day; %, share of all prescribed antibiotics; Δ, difference between 2021 and 2017 DID; Δ %, percentage difference between 2021 and 2017 DID.

Year	2017		2018		2019		2020		2021			
Substance	DID	%	DID	%	DID	%	DID	%	DID	%	Δ	Δ%
All antibiotics	12.581	100.0%	11.925	100.0%	11.242	100.0%	8.786	100.0%	8.056	100.0%	−4.525	−36.0%
Amoxicillin	2.332	18.5%	2.379	19.9%	2.332	20.7%	1.702	19.4%	1.575	19.6%	−0.757	−32.5%
Doxycycline	1.602	12.7%	1.506	12.6%	1.392	12.4%	1.274	14.5%	1.156	14.4%	−0.446	−27.8%
Cefuroxime	2.144	17.0%	1.899	15.9%	1.743	15.5%	1.207	13.7%	0.994	12.3%	−1.150	−53.7%
Amoxicillin and beta-lactamase inhibitor	0.576	4.6%	0.645	5.4%	0.748	6.6%	0.704	8.0%	0.753	9.3%	0.177	30.8%
Clindamycin	0.626	5.0%	0.591	5.0%	0.575	5.1%	0.533	6.1%	0.525	6.5%	−0.101	−16.2%
Phenoxymethylpenicillin	0.636	5.1%	0.607	5.1%	0.610	5.4%	0.397	4.5%	0.298	3.7%	−0.338	−53.1%
Metronidazole	0.079	0.6%	0.077	0.6%	0.074	0.7%	0.070	0.8%	0.069	0.9%	−0.010	−13.0%
Other antibiotics	4.586	36.5%	4.221	35.5%	3.768	33.6%	2.899	33.0%	2.686	33.3%	−1.900	−41.4%

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
