# Peer review of "Assessment of the Prescriptions of Systemic Antibiotics in Primary Dental Care in Germany from 2017 to 2021: A Longitudinal Drug Utilization Study"

_antibiotics, 2022, doi:10.3390/antibiotics11121723_

Round 1

Reviewer 1 Report

Interesting topic to research. However, there are some things that need to be cleared to make it easier for readers to read.

1. The background is self-explanatory, but please note that there are some parts that do not contain citations, so they tend to be speculative.

2. The results are quite clearly presented both in the description and in the table.

3. The discussion is not deep enough. Could it be explained more why there is a decrease in the prescription of some antibiotics by referring to other studies, both in the research site countries and those seen from other countries.

4. The method is sufficient

5. I did not find any conclusions

Reviewer 2 Report

The study is informative and well-conducted but need a few changes. Please see my comments below

Title: indicate the study type: for example

Assessment of the prescriptions of systemic antibiotics in primary dental care in Germany from 2017 to 2021: A retrospective cohort study

Abstract:

 Against the background of growing resistance change to “Due to increasing antibiotic resistance”

 “The current trend of dental antibiotic prescriptions 14 and of clindamycin prescriptions in particular” clindamycin is also antibiotic so change it to “the aim is to determine the trend of antibiotic prescription in primary dental care

Your abstract should have background, method, results, and conclusion
